# Effect of Diabetes Education Through Pattern Management on Self-Care and Self-Efficacy in Patients with Type 2 Diabetes

**DOI:** 10.3390/ijerph16183323

**Published:** 2019-09-09

**Authors:** Sung-Kyoung Lee, Dong-Hyun Shin, Yong-Hyun Kim, Kang-Sook Lee

**Affiliations:** 1Graduate School of Public Health, The Catholic University of Korea, 222 Banpo-daero, Seocho-gu, Seoul 06591, Korea; 2Department of Endocrinology Bundang Jesaeng Hospital 180 Seahyen-ro, Bundang-gu, Gyeonggido 13590, Korea; 3Department of Preventive Medicine, College of Medicine, The Catholic University of Korea, 222 Banpo-daero, Seocho-gu, Seoul 06591, Korea

**Keywords:** Diabetes education, self-care, self-efficacy, type 2 diabetes mellitus

## Abstract

This study investigated the effect of applying a customized diabetes education program through pattern management (PM), using continuous glucose monitoring system (CGMS) results, on individual self-care behaviors and self-efficacy in patients with type 2 diabetes mellitus. Patients with type 2 diabetes who had never received diabetes education, enrolled from March to September 2017, were sequentially assigned to either PM education or control groups. In the PM education group, the CGMS test was first conducted one week before diabetes education and repeated three times by PM in order to obtain data on self-care behaviors and self-efficacy. These results were then compared before and after education at three and six months. The control group received the traditional diabetes education. Self-efficacy showed statistically significant interactions between the two groups over time, indicating a significant difference in the degree of self-efficacy between the PM education and control groups. Diabetes education by PM using CGMS result analysis improved life habits with a positive influence on self-care behaviors and self-efficacy for diabetes management. Further studies are needed to further develop and apply individual diabetes education programs in order to sustain the effects of self-care behaviors and self-efficacy in patients with diabetes who experience a decrease in self-efficacy after three months of education.

## 1. Introduction

Diabetes, one of the four major non-communicable diseases, is defined by the World Health Organization (WHO) as a public health concern with increasing incidence and increasing numbers of patients in the past few decades. According to the global report on diabetes from the WHO, 1,500,000 individuals died from diabetes in 2012, and an additional 2,200,000 people died due to heightened risks of cardiovascular and other diseases from dysglycemia [1]. Regarding deaths due to diabetes among Koreans, the rate decreased by 4.5% between 2006 and 2016—however, diabetes still ranks the fifth among the 10 major causes of death among Koreans, excluding intentional self-harm [2].

Although the development of type 2 diabetes is mostly accounted for by inappropriate life habits, including hypertension, obesity, and hypercholesterolemia, most patients with diabetes do not undertake self-care behaviors, such as dietary changes, exercise, self-monitoring of blood glucose (SMBG), and foot care. Moreover, many patients fail to recognize the importance of continued management and the fact that active, continued self-care behaviors can aid in the prevention of diabetic complications [3]. The diagnosis of type 2 diabetes directly or indirectly affects the patient’s quality of life, and there are reports that patients found to have a negative impact on quality of life have higher HbA1c than those who do not within five years of diagnosis [4].

Recent education programs on diabetes have changed from educator-centered approaches focusing on lectures and information provision to empowerment models in which patients adopt self-care behaviors. In particular, the empowerment models help patients to conduct self-care behaviors that they chose and to actively cooperate with medical staff, while being at the center of diabetic management models [5]. Active behavioral changes in patients constitute the most important aspect of the treatment of diabetes [6]. Rather than using a simple medication therapy, it is more effective in diabetes management to educate patients to understand the disease and to perform self-care behaviors—however, such performance decreases with time after the completion of the education [7,8]. It is important for diabetes educators to develop and provide customized effective diabetes management education by understanding each patient’s conditions. This will promote self-efficacy in self-care behaviors and continued diabetes management [9]. Studies have continued on the development of treatment methods that can prevent diabetic complications and premature deaths, as well as, effective education methods [7,9,10,11,12].

Pattern management (PM) is a systematic process in which records of blood glucose levels are used to understand patterns in the levels, as well as to confirm and analyze factors influencing the levels. By using data collected on the overall life habits, such as blood glucose levels, dietary habits, activity level, and physical and psychological stress, appropriate drug treatment and self-nursing education can be provided to the patients in order to help them improve their ability to self-manage diabetes, and thus to maintain optimal health conditions [13,14]. Data for PM are collected through records of self monitoring of blood glucose (SMBG) or computer-based data collection and management tools. Advancement made in these technologies, which has allowed for timely confirmation of blood glucose patterns, has made it possible for clinicians and patients to make timely decisions in the treatment of diabetes by efficiently utilizing the required information for treatment [14]. Most studies that have applied PM in real-life diabetes education are based on SMBG records [13,15,16]. The frequency of SMBG conducted by clinicians or educators and reported in these previous studies ranged from once to more than seven times. However, it is still necessary to resolve the issues surrounding omitted data. Furthermore, the frequency of SMBG should be determined through the cooperation of clinicians and patients for the optimal clinical use of the data [17].

Continuous glucose monitoring system (CGMS), which was developed to overcome the shortcoming of SMBG, collects glucose data from subcutaneous interstitial fluid. Sensors are first inserted and worn for certain periods, for the measurement; patterns of changes in blood glucose levels can be investigated through data collected every five minutes [18]. Since CGMS records all changes in blood glucose levels that might otherwise be omitted between the intervals of SMBG, it can aid in the effective improvement of glucose levels in diabetes patients, by confirming hyper- or hypoglycemia, and by providing suggestions on appropriate drug treatment, reminders of SMBG, and ways to improve life habits [19]. Through approaches involving education and the improvement of diabetes patients’ blood glucose management, CGMS facilitates cooperative treatment involving patients and medical staff by encouraging open communication between them [20,21]. In fact, many studies have discovered patterns of changes in blood glucose levels in clinical tests through CGMS and confirmed that the application of these patterns to treatments, such as changes in drug regimens or insulin regulation, led to positive changes in glycated hemoglobin (HbA1c) levels [22,23,24]. However, there is lack of research on diabetes education utilizing blood glucose pattern data collected through CGMS and its effects in Korea.

Therefore, the present study aimed to investigate the effects of customized diabetes education through PM, conducted with CGMS results (obtained through individual education programs provided to patients with type 2 diabetes) on individual self-care behaviors and self-efficacy in patients with type 2 diabetes mellitus. This study also aimed to investigate changes in physiological indicators of diabetes in order to determine the basis for the future development of evidence-based individualized diabetes education programs.

## 2. Materials and Methods 

### 2.1. Ethical Consideration

The present study was reviewed and approved by the DMC (Bundang Jesaeng Hospital) institutional clinical research ethical review board (approval number; RN17-01). The work was performed in accordance with the Ethical Principles for Medical Research Involving Human Subjects outlined in the Helsinki Declaration in 1976 (revised in 2000).

### 2.2. Subjects

Patients with type 2 diabetes who visited the Department of Endocrinology at B General Hospital, Gyeonggi, between March 2017 and September 2017 and who provided written informed consent for study participation were enrolled into either the PM education or control group. G Power 3.1 program was used to calculate the sample size required for repeated measures of analysis of variance (ANOVA) with significance level of 0.05, power of 0.95, and effect size of 0.25, resulting in a required total of 44 subjects. Thus, we aimed to include a total of 60 subjects, with 30 in each group.

The subjects were aged between 18 and 70 years, had HbA1c above 8%, and had been receiving treatment for more than six months since the diagnosis, but never had any consultation or education on diabetes. Patients with decreased visual acuity due to diabetic retinopathy or decreased body movement due to diabetic foot diseases were excluded. 

### 2.3. Diabetes Education Program for the PM Education and Control Groups

Both the PM education and control groups received individual education on diabetes and were provided with the Guidelines on Diabetes Management booklet developed by the Korean Association of Diabetes Nurse Educators. The diabetes education program, consisting of two in-person education and one telephonic education sessions, was provided by one endocrinologist, one clinical nutritionist, and one nurse dedicated to diabetes education. All three were certified by the Korean Diabetes Association to provide diabetes education. The first education session was provided by the endocrinologist (on the overview and management goals of diabetes) and by the nurse on SMBG (in both groups, twice daily, on drug therapies as well as prevention and management of chronic complications). The second education session was provided by the clinical nutritionist (on diets) and a nurse (on management in daily life, including prevention and regulation of hypo- and hyperglycemia, exercise, management on days when the patients feel sick, foot care, and stress management). PM was provided to the PM education group. The third education session was provided two weeks after the second session; (whether the patients were performing self-care behaviors, including SMBG, monitoring of hypo- or hyperglycemia and diet improvement, was confirmed, while feedback was provided via telephone calls). 

#### 2.3.1. PM Education Group

For the PM education group, a 60-min primary diabetes education and CGMS test were conducted. The CGMS test was conducted for three days, and CGMS result counseling and individualized PM were provided for 90 min during the second education. PM education was conducted based on a CGMS result graph. For effective self-management, CGMS results were checked in the order of hypoglycemia, fasting hyperglycemia and postprandial hyperglycemia, and suggested improvement directions. In order to improve blood sugar outside the target blood sugar range, the composition of carbohydrates, proteins and fats, and the amount of meals, exercise and the correct use of prescribed medications were checked. For patients, exercise therapy allows them to sweat at least 150 min at least three times a week. A regimen ensures that you eat the right amount, evenly and regularly. During PM education, we suggest that patients actively participate in the improvement activity plan so that they can plan their own lifestyle (meal and exercise). CGMS results and a diabetes education booklet were provided to the patients after the education session. Two weeks after PM management, a third education was conducted by telephone. Surveys to assess self-care behaviors and self-efficacy were conducted immediately prior to the education program, as well as at three and six months after the program.

#### 2.3.2. Control Group

The primary diabetes education was conducted for controls for 60 min. The control group’s primary education program was the same as the PM group, including diabetes management, diet, and exercise management. The control group did not conduct secondary education and only confirmed phone calls after four weeks. Similar to the PM education group, the control group subjects were also provided with a booklet on diabetes education. Surveys to assess self-care behaviors and self-efficacy were conducted immediately prior to the education program, as well as at three and six months after the program.

### 2.4. Study Design and Measurements

This study employed a nonequivalent control group pretest-posttest design to test the effects of PM-based diabetes education utilizing CGMS results of patients with diabetes. The following summarizes the model used in the study design (Figure 1).

#### 2.4.1. General Characteristics of the Subjects 

The general characteristics of the subjects, including sex, age, marital status, educational level, employment, income, height, body weight, duration of diabetes, treatment methods, presence of diabetic complications, and hospital admission, were analyzed.

#### 2.4.2. Self-Care Behaviors

To measure self-care behaviors, the tool used (after obtaining consent from the authors of the tool) was developed by Moon who supplemented and modified the self-care behavior measuring tool, originally developed for patients with diabetes by Choi (1999) [25]. The tool consists of 20 questions, overall: nine diet-related questions, three medication-related questions, three exercise-related questions, and five questions on self-management. The questions were scored on a four-point scale, from one “not at all” to four “very well” and higher scores indicated higher self-care behaviors. The reliability of the tool was reported as Cronbach’s α = 0.91 in Moon’s study (2014). In the present study, Cronbach’s α was 0.94, thus indicating a good reliability. Since the Cronbach’s α of the components ranged between 0.87 and 0.93, the reliability was good overall.

#### 2.4.3. Self-Efficacy

Self-efficacy of the management of diabetes was measured using the tool developed by Song et al. (used by permission of the authors) based on the seven domains of diabetes self-management, suggested by the American Association of Diabetes Educators (AADE, 2008) [26]. The tool, consisting of 17 questions overall, tested the following 6 sub-domains: 2 questions assessed appropriate exercise, two were questions on healthy diet, four were on the monitoring of blood sugar and resolving problems with hypoglycemia, two questions were on the hyperglycemia problem solving, four were on understanding treatment for the prevention of complications, and three questions were on coping with medication and psychological difficulties. All questions were scored on a four-point scale, with the total score ranging between 17 and 68. Higher scores indicated higher levels of self-efficacy. The reliability of the tool was reported as Cronbach’s α = 0.84. In the present study, Cronbach’s α was 0.93, thus indicating a good reliability. Since the components’ Cronbach’s α ranged between 0.66 and 0.88, the reliability was acceptable.

#### 2.4.4. Physiological Index

To evaluate blood glucose management among the subjects before and after the education program, HbA1c levels were determined through high-performance liquid chromatography.

#### 2.4.5. Statistical Analysis

The collected data were analyzed using SPSS (IBM, Armonk, NY, USA) Statistics version 22.0. *X*^2^ tests and independent t-tests were conducted to test for the homogeneity of the subjects’ general characteristics. Self-care, self-efficacy, and HbA1c levels were used for Pearson correlation analysis. Effects on self-efficacy, self-care behaviors, and physiological index were analyzed through repeated measures ANOVA. The significance level was set at < 0.05.

## 3. Results

### 3.1. Test of Homogeneity Between the Groups

#### 3.1.1. General Characteristics of the Study Subjects

There were 60 subjects, overall, with 30 each in the PM and control groups, respectively. The PM education and control groups showed no significant differences in all variables; thus satisfying the requirement for homogeneity of the general characteristics: sex (*p* = 0.184), age (*p* = 0.944), marital status (*p* = 0.688), educational level (*p* = 0.526), monthly income (*p* = 0.218), body weight (*p* = 0.409), height (*p* = 0.988), disease duration (*p* = 0.637), treatment method (*p* = 0.101), presence of complications (*p* = 0.521), and history of admission (*p* = 0.766) (Table 1). 

#### 3.1.2. Self-Care Behavior, Self-Efficacy, and HbA1c

The groups were found to be homogeneous in terms of self-care behaviors, self-efficacy, and HbA1c before the education program: self-care behaviors (*p* = 0.077), self-efficacy (*p* = 0.465), and HbA1c (*p* = 0.334) (Table 1). Self-care behavior showed a significant correlation with self-efficacy (r = 0.833, *p* < 0.001) and Hba1c levels (r = –0.258, *p* < 0.05). Self-efficacy was significantly correlated with HbA1c levels (r = –0.300, *p* < 0.05) (Table 2).

### 3.2. Comparison of Changes in Self-Care Behaviors, Self-efficacy, and HbA1c Between the Groups

#### 3.2.1. Self-Care Behaviors

Between-group differences in self-care behavioral changes showed that interaction between group and time was statistically significant; therefore, there was a significant difference between the PM and control groups in terms of changes in self-care behaviors. With regard to the increase in self-care behaviors, the score increased by 0.38 in the PM group than in the control group three months after the education program. The score decreased by 0.27 in the PM group than in the control group six months after the education program (Figure 2). 

The PM and control groups showed significant mean differences in the four subdomains of self-care behavior. In terms of the interaction between group and time, the PM group demonstrated greater increases than did the control group in the scores for diet-related, exercise-related, and self-management-related questions, three months after the education program; in contrast to the scores for these questions, six months after the education program, the scores decreased in the PM group. Although the scores for medication-related self-care behaviors showed significant mean differences in terms of changes according to time and group, the interaction between time and group was not significant (Table 3).

#### 3.2.2. Self-Efficacy

When group differences in changes in self-efficacy were tested, the interaction between time and group was statistically significant; in other words, the change in self-efficacy demonstrated a statistically significant difference between the PM and control groups. The score for self-efficacy increased much more by 0.51 in the PM group when compared to the control group, three months after the PM-based diabetes education. Six months after the education program, the difference in the reduction between the two groups was only by 0.04, almost identical. In conclusion, self-efficacy increased more in the PM group than in the control group (Figure 3). 

When the interaction between time and group in all six sub-domains of self-efficacy was tested, the scores increased more in the PM group than in the control group (Table 4).

#### 3.2.3. Physiological Index

HbA1c, a physiological index, showed the mean of 9.62 ± 1.25 and 7.72 ± 0.58 before and six months after the education program, respectively, in the PM group. In the control group, the mean HbA1c was 9.69 ± 1.34 before the education program and 8.20 ± 1.1 six months after the program. Since the score decreased by 0.41 in the PM group after the education program, the decrease in HbA1c with time was significant. (Figure 4).

## 4. Discussion

The present study was conducted to determine the basis for the development of customized diabetes education programs by investigating the influence of PM-based diabetes education utilizing CGMS results, on self-care behaviors and self-efficacy of patients with type 2 diabetes.

With changes in self-care behaviors after PM-based diabetes education utilizing CGMS results, the PM group demonstrated improvements compared to the control group—in other words, positive changes in self-care behaviors were observed in the PM group. This is consistent with previous findings that self-care behaviors improved after diabetes education programs [27,28]. Among the sub-domains, medication-related self-care behaviors improved although the difference between groups was not significant. This is similar to previous findings that the experiences of education programs influence medication-related self-care behaviors in lifestyle related diseases. Both groups learned about diabetes and its complications and recognized the necessity of treatment through the diabetes education program; therefore, changes in medication-related self-care behaviors assessed overtime time was not significant [29,30,31].

In this study, changes in self-efficacy before and after training increased by 0.93 for three months before and after PM and decreased by 0.27 after six months of education. The control group increased 0.42, less than half of the PM group after three months of training, and 0.31 after six months of education. Six months after the education program, self-efficacy improved more in the PM group than in the control group, in line with previous findings that diabetes education programs are effective in improving self-efficacy [32,33]. All six sub-domains of self-efficacy showed significant interaction effects of time and group, thus agreeing with the previous findings that diabetes education programs utilizing CGMS results can increase self-efficacy and aid in improving life habits, such as increased physical activities [28,34]. In addition, the study is consistent with previous findings which suggest that self-efficacy improves when diabetics participate in treatment planning, actively learn about the disease, explore the feelings of the disease, and acquire the skills needed to adapt [35]. Diabetes patients with high self-efficacy are known to perform more self-care behaviors, which can potentially prevent diabetic complications and improve the quality of life [36,37]. The present study also showed higher levels of self-care behaviors and self-efficacy in the PM group than in the control group and confirmed the effects of PM utilizing CGMS results.

Regarding continued effects of improved self-care behaviors and self-efficacy after the diabetes education program, the scores decreased three months after the program in both groups as reported in previous studies. However, it is noteworthy in comparison to previous studies that the decrease in continued effects on self-efficacy was significantly lower in the PM group than in the control group [8,38,39,40].

HbA1c, a physiological index, decreased in both the PM and control groups—however, the difference was not significant. This could have resulted from the fact that this present study (conducted to determine PM effectiveness in patients with diabetes), was unable to take into account changes in medication after the education program and differences in treatment methods. Despite the small changes, our results were consistent with the results of many previous studies that observed a decrease in HbA1c after the diabetes education program [41]. 

Based on the results of this study, self-efficiency in diabetes management could be enhanced by directly identifying changes in blood sugar levels, that had not been seen in SMBG, thus improving the coping skills. According to our results, self-efficacy in diabetes management could be enhanced by identifying changes in blood sugar levels that were not identified on SMBG; this would improve the patients’ coping ability. Furthermore, promotion of self-efficacy led to self-care behaviors, which in turn led to effective blood sugar management among diabetes patients.

This study is important in that it could show the effectiveness of PM-based diabetes education on self-care behaviors and self-efficiency after six months, mainly among many Koreans with diabetes, with observed short-term effects after education.

This is especially important because previous studies conducted in Korea have mostly focused on the effects of treatment utilizing CGMS results, particularly changes in physiological indices. This study confirmed that PM-based diabetes education utilizing CGMS results improved life habits by enhancing self-care behaviors and that it exerts positive influences on the promotion of self-efficacy regarding diabetes management. Patients and educators naturally form close relationships while they confirm changes in life habits according to changes in blood glucose patterns during PM-based diabetes education. Furthermore, education also promotes motivation and self-efficacy for self-management of diabetes; this, in turn, encouraged the patients to actively participate in planning ways to improve their life habits.

However, since CGMS is expensive, it was difficult to apply it to all patients with diabetes. Therefore, the study was designed as a small-scale study conducted on patients visiting the department of endocrinology at a general hospital, and it is thus difficult to generalize the results.

Sensors that allowed for the collection of numerous data using CGMS, and that can be used for seven days, were made available close to the end of the study and are being used presently. 

## 5. Conclusions

Diabetes education by PM, using CGMS result analysis, improved the life habit with a positive influence on self-care behaviors and self-efficacy for diabetes management. Along with these changes, the results of this study are expected to serve as the basis for the development of customized diabetes education programs that are specific and individualized, according to each patient’s characteristics through CGMS result analysis. Future studies should develop customized diabetes education programs to sustain continued improvements in self-efficacy and self-care behaviors—these tend to decrease with time following the completion of education programs in patients with diabetes. Furthermore, follow-up studies should be carried out to apply the developed program. The present findings are expected to serve as evidence for changes in policy, including public insurance coverage for CGMS. Furthermore, for CGMS, which can serve as a tool for diabetes education programs, to be more widespread, studies evaluating the cost-effectiveness of CGMS are required.

## Figures and Tables

**Figure 1 ijerph-16-03323-f001:**
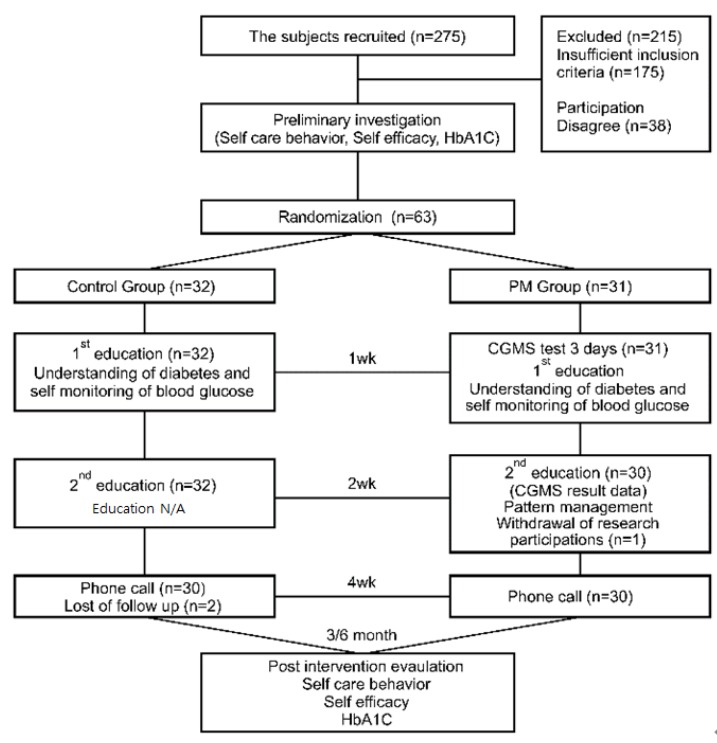
Study framework. CGMS, Continuous Glucose Monitoring System.

**Figure 2 ijerph-16-03323-f002:**
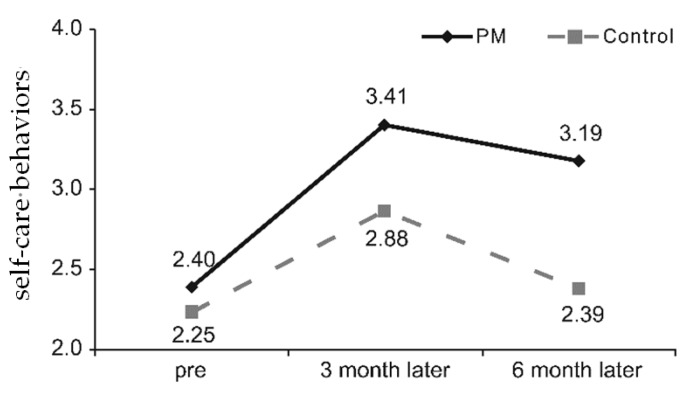
Differences in self-care behavior according to education groups. PM, pattern management.

**Figure 3 ijerph-16-03323-f003:**
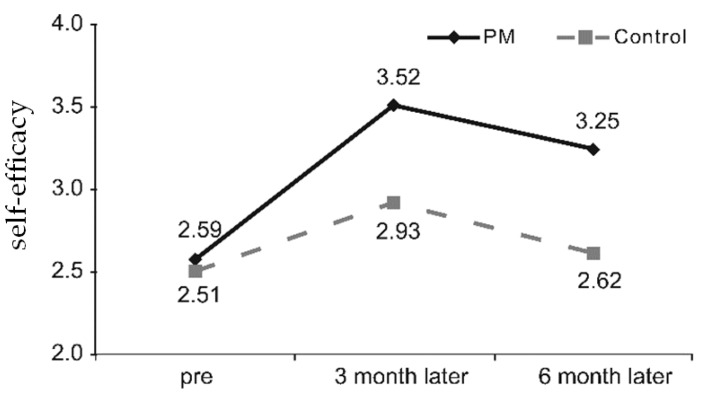
Differences in self-efficacy behavior according to education groups. PM, pattern management.

**Figure 4 ijerph-16-03323-f004:**
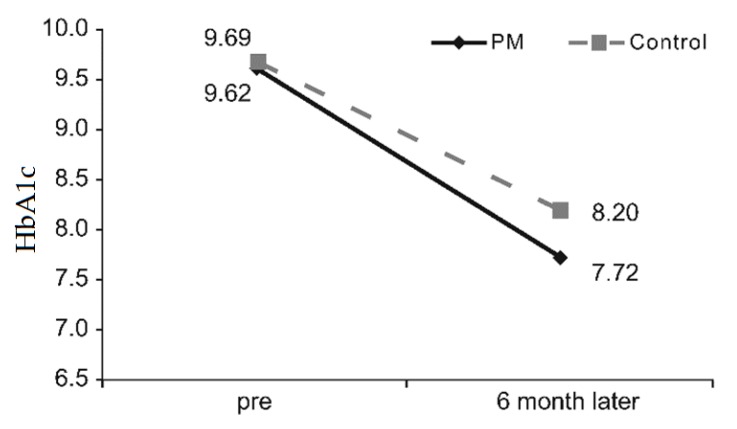
Differences in HbA1c according to education groups. PM, pattern management.

**Table 1 ijerph-16-03323-t001:** General characteristics according to group.

**Variables**	**PM Group** **(*n* = 30)**	**Control Group** **(*n* = 30)**	***χ^2^* or *t***	***p*-Value**
Sex				
Male	16 (53.3)	21 (70.0)	1.763	0.184
Female	14 (46.7)	9 (30.0)		
Age	53.77 ± 9.22	53.60 ± 9.04	0.071	0.944
Marriage				
Married	26 (86.7)	27 (90.0)	0.162	0.688
Single	4 (13.3)	3 (10.0)		
Education				
≤High school	19 (63.3)	19(63.3)	2.232	0.526
≥College	11 (36.7)	11 (36.7)		
Income (10,000 won)				
Low	7 (23.3)	4 (13.3)	5.762	0.218
Moderate	6 (20.0)	10 (33.3)		
High	17 (56.7)	16 (53.3)		
Weight	69.36 ± 13.13	66.87 ± 9.85	0.832	0.409
Height	165.73 ± 10.21	165.77 ± 6.56	–0.015	0.988
Duration(y)	8.73 ± 5.74	9.53 ± 7.25	–0.474	0.637
Treatment				
OHA	4 (13.3)	10 (33.3)	4.591	0.101
Insulin	4 (13.3)	1 (3.3)		
OHA +Insulin	22 (73.3)	19 (63.3)		
Complication				
No	23 (76.7)	21 (70.0)	4.202	0.521
Yes	7 (23.3)	9 (30.0)		
Admission				
Yes	8 (26.7)	7 (23.3)	0.089	0.766
No	22 (73.3)	23 (76.7)		
**Homogeneity of self-care, self- efficacy, and HbA1c according to education groups (mean ± SD)**				
Self-care behavior	2.40 ± 0.33	2.25 ± 0.35	1.798	0.077
Self-efficacy	2.59 ± 0.36	2.51 ± 0.40	0.736	0.465
HbA1c	9.56 ± 1.22	9.94 ± 1.80	–0.976	0.334

OHA, Oral hypoglycemic agents; PM, pattern management; SD, standard deviation.

**Table 2 ijerph-16-03323-t002:** Correlations among self-care behavior, self-efficacy, and HbA1c levels.

Variables	Self-Care Behavior	Self-Efficacy	HbA1c
Self-care behavior	1		
Self-efficacy	0.833 ***	1	
HbA1c	–0.258 *	–0.300 *	1

* *p* < 0.05, *** *p* < 0.001.

**Table 3 ijerph-16-03323-t003:** Difference of self-care behavior factors according to group.

Variables	Group	M ± SD			F		*p*
		Pre	3 Month Later	6 Month Later			
Dietry related	CGMS	2.20 ± 0.37	3.26 ± 0.42	2.97 ± 0.47	Time	118.591 ***	<0.001
	Control	2.13 ± 0.42	2.77 ± 0.48	2.28 ± 0.42	Group	20.427 ***	<0.001
					Time × group	16.600 ***	<0.001
Medication related	CGMS	3.03 ± 0.51	3.83 ± 0.34	3.64 ± 0.49	Time	35.584 ***	<0.001
	Control	2.62 ± 0.91	3.24 ± 0.57	2.88 ± 0.65	Group	23.525 ***	<0.001
					Time × group	2.189	0.128
Exercise related	CGMS	2.12 ± 0.81	3.37 ± 0.56	3.12 ± 0.78	Time	42.969 ***	<0.001
	Control	2.40 ± 0.84	2.89 ± 0.84	2.26 ± 0.69	Group	4.716 *	0.034
					Time × group	19.372 ***	<0.001
Self-management related	CGMS	2.55 ± 0.46	3.47 ± 0.38	3.34 ± 0.42	Time	53.951 ***	<0.001
	Control	2.13 ± 0.61	2.85 ± 0.57	2.38 ± 0.53	Group	54.773 ***	<0.001
					Time × group	5.671 **	0.009

* *p* < 0.05, ** *p* < 0.01, *** *p* < 0.001.

**Table 4 ijerph-16-03323-t004:** Difference of self-efficacy factors according to group.

Variables	Group	M ± SD			F		*p*
		pre	3 Month Later	6 Month Later			
Proper exercise	CGMS	2.58 ± 0.63	3.38 ± 0.57	3.13 ± 0.67	Time	20.526 ***	<0.001
	Control	2.83 ± 0.62	3.00 ± 0.60	2.75 ± 0.47	Group	1.840	0.180
					Time × group	11.743 ***	<0.001
A healthy diet	CGMS	2.62 ± 0.50	3.48 ± 0.58	3.22 ± 0.49	Time	32.511 ***	<0.001
	Control	2.57 ± 0.60	3.02 ± 0.53	2.73 ± 0.41	Group	12.086 **	0.001
					Time × group	4.481 *	0.019
Monitoring of blood sugar and resolving problems with hypoglycemia	CGMS	2.64 ± 0.54	3.63 ± 0.40	3.36 ± 0.57	Time	50.277 ***	<0.001
	Control	2.39 ± 0.63	2.93 ± 0.47	2.57 ± 0.50	Group	32.510 ***	<0.001
					Time × group	7.126 **	0.003
Hyperglycemia problem solving	CGMS	2.05 ± 0.81	3.38 ± 0.57	3.22 ± 0.55	Time	66.011 ***	<0.001
	Control	1.98 ± 0.72	2.83 ± 0.77	2.30 ± 0.52	Group	15.217 ***	<0.001
					Time × group	9.654 ***	<0.001
Understanding treatment for prevention of complications	CGMS	2.81 ± 0.52	3.65 ± 0.42	3.36 ± 0.52	Time	45.116 ***	<0.001
	Control	2.66 ± 0.49	3.06 ± 0.40	2.78 ± 0.35	Group	23.849 ***	<0.001
					Time × group	7.316 **	0.002
Coping with medication and psychological difficulties	CGMS	2.56 ± 0.53	3.40 ± 0.40	3.10 ± 0.51	Time	21.420 ***	<0.001
	Control	2.59 ± 0.56	2.71 ± 0.35	2.50 ± 0.47	Group	22.716 ***	<0.001
					Time × group	14.206 ***	<0.001

* *p* < 0.05, ** *p* < 0.01, *** *p* < 0.001.

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
