# Peer review of "Effect of Diabetes Education Through Pattern Management on Self-Care and Self-Efficacy in Patients with Type 2 Diabetes"

_ijerph, 2019, doi:10.3390/ijerph16183323_

Round 1

Reviewer 1 Report

Interesting study examining self efficacy and self care behaviors in patients with Type 2 diabetes follwoing an education pattern.

Background content was sufficient however more studies identifying quality of life of type 2 pateints and comparison of specialized cohorts (ex. seniors) would  enhance the background.

Methodology: exclusion crieteria included diabetic retinopathy and foot complications which can affect activity. However, discussion of the methodology was confusing in the description of 3 education sessions which did not match the figure. The description of the PM education was not clear as well.

Results: Authors showed results well in a graph but y axis labelling was missing and explaination of the graphs was not clear

Discussion: discussion of the continued effects on self efficacy  in PM group vs control was not clear 

Author Response

Thank you for your kind review

Point 1) Background content was sufficient however more studies identifying quality of life of type2 pateints and compareison of specialized cohorts(ex:seniors) would enhance the background

Response 1) We added the following to the background:.

 (add_line45) “Diagnosis of type 2 diabetes directly or indirectly affects the patient's quality of life, and there are reports that patients found to have a negative impact on quality of life have higher HbA1c than those who do not within 5 years of diagnosis[4]”

Point 2) Methology: exclusion criteria included diabetic retinopathy and foot complications which can affect activity. However, discussion of the methodology was confusing in the description of 3 education sessions which did not match the figure The description of the PM education was not clear as well.

Response 2) We deleted and corrected some of the methodology's sentences. Added a detailed description of PM training to the methodology and revised Figure 1.

(before) PM Education Group

For the PM education group, a 60 - minute primary diabetes education and CGMS test were conducted. The CGMS test was conducted for 3 days, and CGMS result counseling and individualized PM were provided for 90 minutes during the second education. PM education was conducted based on a CGMS result graph. Hypoglycemia was checked on the CGMS result graph to suggest improvement direction. Check for diarrhea or fasting hyperglycemia was done to identify the composition of the diet and suggest directions for improvement. Finally, the postprandial hyperglycemia was checked to identify the composition and the amount of exercise and suggest appropriate improvement directions. Through PM education, we were able to suggest and plan ways to improve the lifestyle (eating and exercise) of our people by encouraging active participation of the subjects. CGMS results and a diabetes education booklet were provided to the patients after the education session. Surveys to assess self-care behaviors and self-efficacy were conducted immediately prior to the education program, as well as at 3 and 6 months after the program.

Control Group

The primary diabetes education was conducted for controls for 60 minutes. The control group's primary education program was the same as the PM group, including diabetes management, diet, and exercise management. Similar to the PM education group, the control group subjects were also provided with a booklet on diabetes education. Surveys to assess self-care behaviors and self-efficacy were conducted immediately prior to the education program, as well as at 3 and 6 months after the program.

(after_line136) For effective self-management, CGMS results were checked in the order of hypoglycemia, fasting hyperglycemia and postprandial hyperglycemia, and suggested improvement directions. In order to improve blood sugar outside the target blood sugar range, the composition of carbohydrates, proteins and fats, and the amount of meals, exercise and the correct use of prescribed medications were checked. For patients, exercise therapy allows them to sweat at least 150 minutes at least three times a week. Regimen ensures that you eat the right amount, evenly and regularly. During PM education, we suggest that patients actively participate in the improvement activity plan so that they can plan their own lifestyle (meal and exercise).

(add_line149)

Two weeks after PM management, a third education was conducted by telephone.

(add_line156)

The control group did not conduct secondary education and only confirmed phone calls after four weeks.

(before_ figure1)                                          (after_ figure1)

Point 3) Results: Authors showed results well in a graph but y axis labeling was missing and explaination of the frapjs was not clear.

Response 3): We have labeled the y-axis and added a statistical table to clarify the graph.

(before_figure2)

(before_figure3)

(before_figure3)

(after_figure2)

(after_figure3)

(after_figure4)

(add_line 242, table1)

Table 1. Difference of self-care behavior factors according to group

Variables

Group

M±SD

F

p

pre

3 month later

6 month later

Dietry related

CGMS

2.20±0.37

3.26±0.42

2.97±0.47

Time

118.591***

<0.001

Control

2.13±0.42

2.77±0.48

2.28±0.42

Group

20.427***

<0.001

Time×group

16.600***

<0.001

Medication related

CGMS

3.03±0.51

3.83±0.34

3.64±0.49

Time

35.584***

<0.001

Control

2.62±0.91

3.24±0.57

2.88±0.65

Group

23.525***

<0.001

Time×group

2.189

.128

Exercise related

CGMS

2.12±0.81

3.37±0.56

3.12±0.78

Time

42.969***

<0.001

Control

2.40±0.84

2.89±0.84

2.26±0.69

Group

4.716*

.034

Time×group

19.372***

<0.001

Self-management related

CGMS

2.55±0.46

3.47±0.38

3.34±0.42

Time

53.951***

.<0.001

Control

2.13±0.61

2.85±0.57

2.38±0.53

Group

54.773***

<0.001

Time×group

5.671**

.009

* p<.05, ** p<.01, *** p<.001

(add_line258, table2)

Table 2. Difference of self-efficacy factors according to group

Variables

Group

M±SD

F

p

pre

3 month later

6 month later

Proper exercise

CGMS

2.58±0.63

3.38±0.57

3.13±0.67

Time

20.526***

<0.001

Control

2.83±0.62

3.00±0.60

2.75±0.47

Group

1.840

.180

Time×group

11.743***

<0.001

A healthy diet

CGMS

2.62±0.50

3.48±0.58

3.22±0.49

Time

32.511***

<0.001

Control

2.57±0.60

3.02±0.53

2.73±0.41

Group

12.086**

.001

Time×group

4.481*

.019

Monitoring of blood sugar and resolving problems with hypoglycemia

CGMS

2.64±0.54

3.63±0.40

3.36±0.57

Time

50.277***

<0.001

Control

2.39±0.63

2.93±0.47

2.57±0.50

Group

32.510***

<0.001

Time×group

7.126**

.003

Hyperglycemia problem solving

CGMS

2.05±0.81

3.38±0.57

3.22±0.55

Time

66.011***

<0.001

Control

1.98±0.72

2.83±0.77

2.30±0.52

Group

15.217***

<0.001

Time×group

9.654***

<0.001

Understanding treatment for prevention of complications

CGMS

2.81±0.52

3.65±0.42

3.36±0.52

Time

45.116***

<0.001

Control

2.66±0.49

3.06±0.40

2.78±0.35

Group

23.849***

<0.001

Time×group

7.316**

.002

Coping with medication and psychological difficulties

CGMS

2.56±0.53

3.40±0.40

3.10±0.51

Time

21.420***

<0.001

Control

2.59±0.56

2.71±0.35

2.50±0.47

Group

22.716***

<0.001

Time×group

14.206***

<0.001

* p<.05, ** p<.01, *** p<.001

Point 4) Discussion of the continued effects on self efficacy in PM group vs control was not clear.

Response 4) We added a difference in the persistence of self-efficacy between the PM training group and the control group.

(Before) Six months after the education program, self-efficacy improved more in the PM group than in the control group, in line with previous findings that diabetes education programs are effective in improving self-efficacy [31,32]. All six sub-domains of self-efficacy showed significant interaction effect of time and group, thus agreeing with previous findings that diabetes education programs utilizing CGMS results can increase self-efficacy and aid in improving life habits, such as increased physical activities [27,33]. Diabetes patients with high self-efficacy are known to perform more self-care behaviors, which can potentially prevent diabetic complications and improve the quality of life [34,35]. The present study also showed higher levels of self-care behaviors and self-efficacy in the PM group than in the control group and confirmed the effects of PM utilizing CGMS results.

.

(add_line283) In this study, changes in self-efficacy before and after training increased by 0.93 for 3 months before and after PM and decreased by 0.27 after 6 months of education. The control group increased 0.42, less than half of the PM group after 3 months of training, and 0.31 after 6 months of education.

Reviewer 2 Report

In the Discussion section, it could be highlighted about other countries where education in diabetes is applied, where diabetic patients and their families are educated and specially attended about their disease. This control is once a month to carried probes to control diabetes.

Author Response

Thank you for your kind review

Point 1) In the discussion session, it could be highlighted about other countries where education in diabetets is applied, where diabetic and their families are educated and specially attended about their disease. This control is once a month to carried probes to control diabetes.

Response 1) We add on the basis of a document of a systematic review of existing literature from 1985-2001 that explains the importance of improving self-efficacy and self-management skills in the education of diabetes.

(Before) Six months after the education program, self-efficacy improved more in the PM group than in the control group, in line with previous findings that diabetes education programs are effective in improving self-efficacy [31,32]. All six sub-domains of self-efficacy showed significant interaction effect of time and group, thus agreeing with previous findings that diabetes education programs utilizing CGMS results can increase self-efficacy and aid in improving life habits, such as increased physical activities [27,33]. Diabetes patients with high self-efficacy are known to perform more self-care behaviors, which can potentially prevent diabetic complications and improve the quality of life [34,35]. The present study also showed higher levels of self-care behaviors and self-efficacy in the PM group than in the control group and confirmed the effects of PM utilizing CGMS results.

(add_line291) In addition, the study is consistent with previous findings suggesting that self-efficacy improves when diabetics participate in treatment planning, actively learn about the disease, explore the feelings of the disease, and acquire the skills needed to adapt [35].

This manuscript is a resubmission of an earlier submission. The following is a list of the peer review reports and author responses from that submission.